# Methadone in Swedish specialized palliative care—Is it the magic bullet in complex cancer-related pain?

**Per Fürst**[1,2]*, **Staffan Lundström**[1,2], **Peter Strang**[1,2]

**1** Department of Oncology-Pathology, Karolinska Institutet, Stockholm, Sweden, **2** Palliative Medicine, Stockholms Sjukhem Foundation, Stockholm, Sweden

* per.furst@ki.se

## Abstract

### Context

Despite being associated with dependence and social stigma, methadone is a potential end-of-life option in complex cancer pain.

### Objectives

To explore attitudes and opinions about methadone and its potential role and current use in complex end-of-life pain.

### Methods

Semi-structured interviews (n = 30) with physicians in specialized palliative care, transcribed and analyzed with conventional qualitative content analysis.

### Results

According to the physicians, patients and relatives expressed unexpectedly few negative attitudes, not affecting methadone's use as an analgesic. Complex pain in bone-metastatic cancer of the prostate, breast and kidney, as well as pancreatic cancer and sarcomas were recurrent suggestions of appropriate indications.

Most of the informants stated that they applied a mechanism-based treatment and mainly prescribed low-dose methadone as an add-on to an existing opioid therapy to benefit from methadone´s proposed NMDA-receptor inhibiting properties, e.g. in cases with reduced opioid sensitivity. Despite its complex pharmacokinetics with a long half-life, most informants expressed defined strategies to avoid side-effects such as respiratory depression, especially when initiating treatment in the home-care setting.

While many palliative care physicians expressed an overly enthusiastic attitude, others stressed the risks of overconfidence, low precision in use, and overlooked treatment options. Besides the obvious physical pain-relieving effects, they stated that effective pain relief could result in a reduced workload and emotional empowerment, both for physicians and staff.

**Data Availability Statement:** Data cannot be shared publicly due to the Swedish regulations based on the "Personuppgiftslagen" (Privacy Act), GDPR and the ethical requirements stipulated by EPN (The Regional Ethical Review Board in

Stockholm Region) in their ethical approval. Data are available from the Data Protection Representative responsible for GDPR at the Stockholm Sjukhem Foundation (contact via dso@stockholmssjukhem.se) for researchers who meet the criteria for access to confidential data.

**Funding:** This study was financially supported by the Stockholm Sjukhem Foundation to PS. The funder had no role in study design, data collection and analysis, decision to publish, or preparation of the manuscript.

**Competing interests:** The authors have declared that no competing interests exist.

## Conclusion

Methadone, especially in the form of low-dose add-on to other opioids is widely advocated in Swedish specialized palliative care as a practical and safe method with rapid onset in complex pain situations at the end of life.

## Introduction

### Complex pain

Strong opioids are often first-line treatment options for cancer-related pain. Unfortunately, they may have an insufficient effect on neuropathic pain, mixed nociceptive and neuropathic pain, pain due to inflammation, or on intense pain with peripheral or central sensitization. These complex pain situations therefore constitute a challenge [1–3]. Central sensitization is defined by the IASP as an increased responsiveness of nociceptive neurons in the central nervous system [4]. Normal or subthreshold afferent input can cause an accelerated pain state despite no progressive tissue damage [4–6]. Central sensitization is partially mediated by activation of the N-methyl-D-aspartate (NMDA) receptors in the central nervous system and is characterized by opioid tolerance, development of allodynia, hyperalgesia, and opioid-resistant neuropathic pain [7].

### Methadone

Compared to other opioids, the partially unique analgesic properties of methadone in combination with highly variable pharmacokinetics can contribute to dosing difficulties, with potential risks [8, 9]. Methadone has a predominantly mu-receptor stimulating effect, but also inhibits NMDA-receptors [10–13]. This is of particular interest in complex pain syndromes. In such cases, combining an ongoing high-dose opioid treatment with a low fixed dose of methadone has become increasingly popular in specialized palliative care, not least in Sweden [14–18]. Still, the appropriate indications for, as well as practical uses of, low-dose methadone have been insufficiently defined. Moreover, methadone maintenance therapy associates methadone with opioid use disorder, which can be socially stigmatizing and may influence methadone's use in specialized palliative care, playing a role as barrier to treatment [19]. Despite great advances in pain treatment, especially as regards cancer pain, physicians within specialized palliative care still encounter a significant number of patients with pain that is unresponsive even to high-dose opioid treatment.

In 2018, approximately a tenth of the 92,185 individuals who died in Sweden were enrolled in specialized palliative care [20].

### Aim

The aim of the study was to explore negative as well as positive attitudes and opinions regarding methadone, including its potential role and current use, in complex pain in specialized palliative care in Sweden.

## Methods

As there is limited knowledge about the studied phenomenon, a qualitative, explorative approach was chosen and reported in accordance with the Consolidated criteria for reporting qualitative research (COREQ) [21].

## Research team

**Personal characteristics.** All three researchers were consultant physicians in palliative medicine. The interviewing researcher was a doctoral student focusing on methadone use in palliative care and the others were associate professor and professor, respectively.

**Relationship with participants.** Some of the informants and the researchers had professional knowledge of each other, whereas the others were previously unknown colleagues within the community of Swedish palliative medicine.

## Study design

For this study, we refrained from using a theoretical framework. Instead, a qualitative conventional content analysis, as described by Hsieh et al., was used [22]. This type of design is usually appropriate when there is limited literature or theory available on a phenomenon.

**Participant selection.** In order to catch as many aspects of the phenomenon as possible, participants were selected through purposeful maximum variation sampling regarding age, gender, geographical representation, level of education and experience of palliative medicine or pain medicine. The participants were approached by email and were either physicians working in specialized multi-professional palliative care teams in Sweden in palliative home care or on palliative care wards, or pain specialists. No participants of those approached refused to participate or dropped out.

**Setting and data collection.** Data were collected in private at the participants workplaces by audio-recorded semi-structured interviews, followed by verbatim transcription. The interview guide, initially pilot tested on two participants, included questions regarding the participants' attitudes towards methadone, and their professional experiences of using it for pain control. Examples of initial open-ended questions were: "Describe any occurring prejudices about methadone that you have encountered [in colleagues, staff, patients or their next of kin]", "Tell me about your experience of treating pain over the years", and, for physicians positive to methadone use: "Describe a patient who you would expect to benefit from the use of methadone for analgesia". When needed, follow-up questions were posed, further questions added, and new emerging areas of interest were explored. All interviews were performed by the first author and lasted for 30–55 minutes. The number of participants was determined based on "saturation", which occurs when continuing data collection provides no further substantial information and when patterns in data become evident. Saturation was mainly reached after 15–20 interviews but, as a safeguard, a total of 30 informants were interviewed. The interviews were conducted mainly from November 2017 to February 2018.

## Data analysis

A qualitative conventional content analysis was performed, according to the steps and nomenclature as described by Hsieh et al. [22] by two of the authors as follows: To become acquainted with the content of the interviews, they were read through several times. They were then read word by word to identify meaning units, i.e. word or text segments, patterned meaning, and issues of potential interest with reference to the research question. The short segments were labeled with preliminary codes. Based on meaningful differences and similarity, codes were brought together and sorted into clusters, i.e. preliminary categories. The preliminary categories were closely examined and compared to find their essence, and then further combined into four final categories which constituted the manifest content. These final categories were discussed by the researchers and, when needed, revised in order to obtain agreement. The software OpenCode 4.0 was used to manage data [23].

## Trustworthiness

In qualitative studies, trustworthiness implies giving the most probable meaning from a particular perspective and can be described using the concepts: credibility, that is, how data and the analysis address the intended aim; dependability, how data change over time and about alterations in decisions during the analysis process; and transferability, which refers to how the results could be transferred to another context [24]. Trustworthiness was ensured by an ongoing process of reflection on these concepts. During the interviews, similar questions were posed in different ways to ensure that the informant's view was correctly captured [25]. To reduce the risk of investigator bias, the authors individually analyzed relevant parts of interviews and compared their findings [26]. To further strengthen credibility, an intercoder concordance was calculated. The supervising author coded blindly 30 citations using the categories set by the first author. In 29/30 cases (97%), the supervising author chose the same code as the first author. In 5 of the longer citations (17%), the supervising author double-coded the citation, i.e. allocated it also to an additional category.

Also, the authors contested and supplemented each other's readings [27]. Common descriptions were formulated, rather to find possible alternative interpretations than aiming at reaching a consensus [27]. To enhance transferability and to illustrate the categories, representative quotations from the transcript as examples of explicated meanings were provided [28].

## Ethical considerations

The study was approved by the Regional Ethical Review Board in Stockholm (2017/230-32). All the participants accepted to participate in the study and written informed consent was obtained.

## Findings

For characteristics of the informants, see Table 1. Main categories and subcategories are summarized in Table 2. Quotations are inserted in the text.

The initial focus of the study was to explore negative as well as positive attitudes and opinions regarding methadone. However, it soon became apparent that negative attitudes were few among physicians. In fact, most of the informants were more enthusiastic about, and provided spontaneous aspects on, the benefits and practical uses of methadone.

## Attitudes

Attitudes towards methadone varied from skepticism to an overwhelmingly positive feeling. In this aspect, the physicians tried to distinguish between their own perceptions, the attitudes of the staff, and the perceived attitudes of the patients and their relatives. Although the primary attitude in general was positive, there were a few situations where negative attitudes among doctors, patients or relatives had constituted a barrier for the initiation of treatment, as well as situations where patients felt uncomfortable.

*People often frown upon the suggestion of it [methadone] . . .//. . . they associate it with substance use disorder. (Informant 16)*

**Physicians and staff.** Enthusiasts among palliative care physicians described methadone as the magic bullet, whereas other informants were mildly positive but also placed an emphasis on the risks of being naively positive: the risk of unselected use with low precision, the risk of

**Table 1. Characteristics of 30 physicians working in specialized palliative care or pain medicine, interviewed in Sweden.**

|  | *Number (%)* |
|---|---|
| **Mean age (range), years** | 53 (26–68) |
| **Female** | 22 (73) |
| **Male** | 8 (27) |
| **Years as physician, median (range)** | 24 (8 months-40 years) |
| 0–10 | 1 (3) |
| 11–20 | 11 (37) |
| 21–30 | 8 (27) |
| >30 | 10 (33) |
| **Years in palliative care/pain medicine, median (range)** | 8 (8 months-30 years) |
| 0–5 | 13 (43) |
| 6–10 | 8 (27) |
| 11–20 | 6 (20) |
| >20 | 3 (10) |
| **Basic specialty** | |
| Oncology | 9 (30) |
| Family Medicine | 8 (27) |
| Internal Medicine | 4 (13,5) |
| Anesthesiology | 4 (13,5) |
| Geriatrics | 3 (10) |
| Gynecology | 1 (3) |
| Under training | 1 (3) |
| **Additional specialty** | |
| Palliative Medicine [1] | 3 |
| Nephrology | 2 |
| Pain Medicine | 1 |
| Hematology | 1 |

[1] Palliative medicine has been a medical add-on specialty in Sweden since May 2015. At least 2.5 years of training is required to apply for this specialty.

**Table 2. Main categories and subcategories.**

| **Attitudes** |
|---|
| Physicians and staff |
| Attitudes among patients and relatives |
| **Indications** |
| Mechanism-based prescription |
| End-of-life situations with short life expectancy |
| **Practical use** |
| Low-dose add-on strategy |
| Effects and adverse effects |
| Initiation, settings |
| The dying patient |
| **Refractory pain situations** |

overlooking other targeted options, as well as the problems with a complex pharmacology and the possibility of severe adverse effects such as respiratory depression.

> *[Pain specialist:] When methadone is introduced, it sometimes implies an overuse of an unconsidered treatment, while there probably sometimes are [alternative] oncological treatments, or it is more proper to consider a spinal catheter or something else. (Informant 14)*

Less experienced physicians had a positive attitude but were less prone to initiate treatment unless they had an experienced colleague to lean on. Also, according to the physicians, palliative care nurses with experience of patients on methadone had a positive attitude and regularly asked the doctor whether methadone would be an option in complex situations.

Besides the obvious pain relief resulting from methadone treatment, the informants also emphasized the effects of effective pain treatment on quality of life. Successful pain control transformed a chaotic situation to a peaceful one, family members were happy, and freedom of pain fostered hope in the patients with opportunities both for recuperation and preparation for impending death.

> *Good pain relief. What does it mean for the patients? How long an answer can I give? Better quality of life, opportunities to take care of themselves and fill everyday with things that are important to them. (Informant 17)*

Not only was successful pain relief important for the patients and their families, it also decreased the staffs' and the palliative care physicians' actual work loads and their feelings of powerlessness.

> *And how does this [effective pain relief with methadone] affect you? Obviously, I feel very competent! It makes me very happy! (Informant 25)*

In contrast to the generally positive attitude among the palliative care physicians, some of them also spontaneously depicted what they perceived as a less positive attitude among pain specialists at pain clinics. This opinion was based on their own contacts with pain specialists, regarding specific patients. When asking about their own thoughts about this discrepancy, some of them suggested that pain specialists probably treat other types of patients in other contexts, possibly with other pain mechanisms, but have limited first-hand experience of treating dying cancer patients, as these generally are treated by palliative care specialists. Despite a less positive general attitude among pain specialists, the role of add-on methadone was at least partly acknowledged.

> *[Palliative specialist:] But, I have seen how they [the pain specialists] have struggled with the same patients that we have later achieved analgesia for. No, we usually do not involve them, only when intrathecal catheters are required. Otherwise not, since we have a better understanding of these things [=in this special context]. I know—I am not being very humble [in this matter]. (Informant 4)*

> *[Pain specialist:] Sometimes you see a remarkably good short-term effect of a low dose [methadone] as add-on. But I don't think the effect will last.. . .//. . .In the selected group of patients that I meet, it is extremely rare that you see any differences [in pain] when you discontinue methadone treatment [after a longer period of treatment], it makes no difference. (Informant 14)*

**Attitudes among patients and relatives.** According to the physicians, there were preconceptions among patients and relatives. They pictured a general hesitance among patients for

strong opioids, but methadone was particularly associated with substance dependence, especially with heroin. Collecting the medications at the local pharmacy was associated with a concern that others may think the individual was a person with a substance abuse disorder. This was seldom a serious obstacle, however. Patients who were initiated on methadone with a good pain-relieving effect were especially prone to continue the treatment. However, persons with former or current opioid use disorder and with cancer were in general skeptical towards methadone and frequently asked for an alternative opioid.

## Indications

In specialized palliative care, methadone was regularly delineated as an important tool for advanced pain treatment, especially in complex malignant pain with long duration, but also in situations with an undesirable, rapid escalation of opioid doses. Patients with bone pain from metastases and pathologic fractures were recurrent examples. Among specific cancer diagnoses, prostate, breast and renal cancer with bone metastases, as well as sarcomas, were typical emerging examples. As regards localizations, the spine and the pelvis but also peritoneum (carcinosis) were recurrently mentioned by most of the physicians.

*Describe a typical patient for whom you think methadone would reduce pain?*

*A patient with widely disseminated bone metastases and mixed [nociceptive and neuropathic] pain components. What I aim at is . . .//. . . a complex pain situation with a patient on high [opioid] doses. . .//. . .who probably had radiotherapy, but where we did not fully succeed [with pain relief]. (Informant 15)*

**Mechanism-based prescription.**   Except for a few of the less experienced physicians, most stated that their pain treatment strategies in individual cases were based on a pain analysis in order to identify clinical pain mechanisms and pain types (e.g. nociceptive, neuropathic or mixed pain). In the case of mixed nociceptive and neuropathic pain, which in cancer patients was perceived to be much more common than pure neuropathic pain, methadone was often used as an add-on to ongoing high-dose opioid treatment. The effect of this strategy was sometimes described as unexpectedly good.

*First you just raise the basic dose of opioids, but when you start coming up in dose and it does not help then I would immediately say to try methadone, just because it influences neuropathic pain . . .//. . . It is fantastic, I could sell it. (Informant 4)*

The participants listed several mechanism-based considerations where methadone was one of the treatment options, e.g. wind-up mechanisms, central sensitization, "exhausted" pain system, "receptor fatigue" and opioid tolerance. Informants perceived that approximately 50% of the selected group of patients that were initiated on methadone tended to respond, but response rates of up to 90% were mentioned.

*Some patients improve. They get a better pain effect than you would expect from five milligrams, if you see what I mean. (Informant 15)*

Compared to ketamine, the effect of methadone on neuropathic pain was described as similar. However, the possibility to also prescribe methadone in the earlier stages of a palliative trajectory and in the form of tablets was highlighted.

**End-of-life situations with short life expectancy.**   Methadone as a low-dose add-on treatment was considered to play an important role in specialized palliative care in end-of-life situations where a rapid effect is needed, as the routines for initiation and dose-escalation are rather simple.

*[I add low-dose methadone] when I don't have time to wait, when gabapentin or other [anti-neuropathic] drugs are too slow [for the patient], i.e. in more urgent situations. (Informant 26)*

*But if there is just a short life expectancy, I use [low-dose] add-on [methadone]. (Informant 26)*

The informants also highlighted patients with an impaired renal function as well as elderly patients as suitable candidates for methadone treatment in comparison to morphine treatment.

## Practical use

In general, most of the practical aspects were about initiation and dosing, this in order to avoid severe adverse effects, among which respiratory depression was the most feared. Additionally, special settings such as home care and initiation in imminently dying patients were highlighted.

**Low-dose add-on strategy.**   Although a few physicians did use methadone as a primary opioid, most of the informants were only familiar with low-dose treatment as an add-on to an already existing, insufficient, high-dose treatment with strong opioids. In such situations, the aim was to provide an NMDA-receptor inhibition to overcome, e.g. opioid tolerance or to dissolve a situation with central sensitization.

*So, it [low-dose add-on methadone] gives an NMDA-blockade which, theoretically, sounds a very smart way of reducing sensitization etc. (Informant 21)*

Even though the physicians worked in different services and in different parts of the country, their dosing and strategy was similar: most of them started the peroral treatment with 2.5 mg 2–3 times/day. A dose escalation was performed after two to seven days, and in most cases, they settled for a dose of 5 mg b.d. and rarely more than 10 mg b.d. as they stated that the improvement of the effect culminated with such doses. This contrasted with the treatment strategy when methadone was used as a primary opioid aimed at the mu-receptor effect. In these cases, a higher daily dose than 20 mg was regularly needed and, instead of b.d., the preferred long-term dosing was in many cases three times per day. In parallel with dose escalation, the physicians had concerns about the risk of cardiac arrythmias due to QT-prolongation.

**Effects and adverse effects.**   In successful cases, the pain-relieving effect set in already within hours or up to a day and was enforced in parallel with the dose escalation. When pain was reduced, adverse effects, especially sedation, started to occur. Whereas less experienced physicians suggested a reduction of the methadone dose in such instances, the more experienced physicians interpreted this as an opioid side-effect caused by the already existing high-dose treatment with standard opioids, rather than by the low-dose methadone. They therefore had a routine to reduce the primary high-dose opioid by about 25–30% as soon as a pain-relieving effect was registered. Nurses were instructed to monitor sedation and breathing patterns although, ultimately, the monitoring of adverse effects was a doctor´s responsibility. The most experienced physicians were also aware of the risk of suddenly emerging opioid adverse

effects if the pain had been managed by other means, e.g. by radiotherapy for bone pain—when pain fades away, opioid adverse effects become more prominent.

> *If you have a pain that is almost not morphine sensitive at all, you can increase to very high opioid doses and still have a normal respiration [as pain is a strong stimulus for the brain stem]. If you then add something [addressing another pain mechanism] that makes you pain-free, you still have the high [opioid] dose and the whole problem [with adverse effects] emerges. (Informant 30)*

Effects and adverse effects appeared in a given sequence according to the informants, given that the dose of the high-dose opioid was not reduced. First, there was an apparent pain reduction, followed by cognitive adverse effects, thereafter sedation and, eventually, respiratory depression. In general, such adverse effects were registered 2–4 days after initiation of the treatment.

In contrast to methadone use as a primary opioid with a long-lasting effect, most of the respondents pictured an effect limited in time when using methadone as an add-on strategy to obtain an NMDA-inhibition. Typically, the effect was described in terms of weeks to a few months. However, this was enough in most cases, as they were treating terminally ill patients.

> *Have you noticed if the analgesic effect you want [by adding low-dose methadone] persists over time? "Yes, for quite a long time anyway. I think several weeks to a few months. (Informant 29)*

**Initiation, settings.**   If methadone was initiated in a home care setting instead of on a ward, some prerequisite conditions had to be fulfilled, namely, a stable home environment with no substance abuse and no severe psychiatric or social problems. Provided these conditions were met, the team needed to have daily contact during the first 3–5 days in the form of home visits or, in stable situations, by telephone calls. Moreover, both the patient and family were to be aware and observant of possible adverse effects, such as sedation or respiratory depression, and instructed to immediately alert the palliative care service.

**The dying patient.**   As a rule, methadone treatment was initiated with tablets. When the dying patient lost his or her ability to swallow, three options were then possible according to the informants. Some preferred to convert the peroral dose to intermittent subcutaneous administration twice daily using 50% or, occasionally, up to 100% of the peroral dose. Intermittent injections were often preferred even when a syringe driver for subcutaneous use was prescribed for the administration of other essential drugs. However, a few palliative care units had routines for including methadone in the syringe driver, mostly with no registered complications or interactions. In a few cases, peroral methadone administration was terminated without a parenteral substitute.

> *Are there any advantages with the use of methadone in a subcutaneous syringe driver?*

> *Yes, in fact I would say that the option [to use it in a syringe driver] is the primary advantage over all other anti-neuropathic drugs. It means a great deal [as dying patients may need a conversion to parenteral drug delivery]. (Informant 3)*

## Refractory pain situations

Methadone was pictured as a solution in many, but far from all, cases of complex pain problems. According to the informants, complex pain with several pain mechanisms and

concomitant death anxiety was a great challenge. These situations were characterized by components of general anxiety, existential suffering, death anxiety or social pain; ordeals difficult to manage merely with analgesics.

> *Severe pain is a matter of so many different components, not only physical pain. The physical pain itself is often the simplest part, whereas the existential and the psychosocial components are the most difficult. (Informant 16)*

## Discussion

### Attitudes

Methadone prescription is occasionally associated with heroin use, dependence and social stigma which might constitute a practical problem in pain management. The subject has been sparsely studied, but Shah *et al.* reported in 2010 that many clinicians fail to prescribe methadone precisely due to such preconceptions [19]. However, this was not the case in the current study. Most of the informants were enthusiastic and stated with assurance that the attitude was also positive among staff, patients and their families and did not constitute an obstacle to its use for pain management. Instead, the attitude among several palliative care physicians was reported to be overly positive, to a degree where there, probably, was a potential risk of overlooking other treatment options for complex cancer pain. It is noteworthy that the Cochrane review from 2017 on the effects of methadone on neuropathic pain in adults, was inconclusive, mainly due to small and low-quality studies [29].

The reported attitude among pain specialists was often more reserved, possibly, because they seldom encounter these types of pain situations. In Swedish health care, pain specialists are mainly involved in chronic non-cancer pain management as well as in situations where spinal catheters or nerve blocks are needed. Most of the complex pain situations in cancer patients, however, are managed by palliative care specialists. Additionally, if the assumption is made that the NMDA-inhibiting effect is the main mechanism behind successful low-dose add-on treatment, the differing views between palliative care physicians and pain specialists could be interpreted as follows: A pain specialist, typically encountering cancer patients in an earlier palliative stage, might use methadone in complex pain and identify a considerable initial effect (as in the quotation by Informant no 14), but might also, after a longer period of treatment, end the methadone medication, without subsequent increase in pain. The NMDA-inhibition, be it achieved by methadone or ketamine, has a potential to reverse opioid tolerance [30], hence making pain responsive to opioids again (the mu-receptor effect). Methadone can then safely be removed, if the patient is still on other opioid medication. This would explain the quotation by Informant 14, who acknowledged a good initial effect in certain cases, but underlined that it was possible to end methadone treatment later in the course without an increase in pain. In contrast, the palliative care specialist is typically in charge for the last few weeks of life and during this period witnesses the initial NMDA-inhibiting effect with reduced pain levels, contributing to their often overly positive attitude towards the treatment.

### Indications for methadone and refractory pain situations

According to the interviewed palliative care physicians, selected patients could benefit from the use of methadone. Often, not least when time was limited, unexpectedly good and safe analgesia was reported to be achieved with low-dose add-ons to high doses of other opioids, which was also the case in home care. Patients with complex cancer-related pain involving

central sensitization, as well as mixed nociceptive-neuropathic pain arising from bone metasta-ses were delineated as two especially important groups.

In specialized palliative care, treatment of complex pain is one of the most prominent and challenging tasks. Excellent conditions for successful pain treatment can be obtained by first identifying, and then addressing, the mechanism behind the pain, be it nerve pain, inflamma-tion, muscle spasm or an opioid-sensitive pain. [31–34]. Most of the Swedish palliative care physicians in this study said they applied this mechanism-based concept for pain management.

During the cancer trajectory, the demands for rapid and effective pain relief increase. Meth-adone was highlighted as an important component of the palliative toolbox because it was easy to handle and had rapid onset, often within a few hours. The notion of methadone treatment being successful in 50–90% of selected patients is mainly in line with previous reports of the effects of low-dose methadone add-on therapy [15, 18, 35, 36].

However, not only the physical components of pain were of interest. When asked about dif-ficult or refractory pain components, a recurrent answer was pain situations with a lot of anxi-ety/death anxiety or general existential anxiety. This component is only partially responsive to drug treatment, instead, supportive strategies are needed. Although the concept of existential pain is not an approved term by the IASP, the role of existential aspects is included in the DSM-V classification where it is stated that the diagnosis is made based on "distressing somatic symptoms plus abnormal thoughts, feelings and behaviors in response to these symp-toms" [37, 38]. Furthermore, the psychological and existential aspects are firmly rooted in the palliative concept of "total pain". Moreover, the very definition of pain, according to IASP, underlines that pain is "an unpleasant sensory and *emotional* experience. . ."[4]. Existential anxiety, especially in the form of death anxiety is definitively a strong, emotional experience.

## Practical use

In specialized palliative care in Sweden, 96% of patients using methadone receive it in the form of low-dose add-on, above all to take advantage of its presumed NMDA-receptor inhibitory effect [18]. Low-dose add-on was, by our informants, considered a simple and manageable way of obtaining the desired effects. This probably explains the low total doses and the two-dose regimen used. However, when methadone was occasionally used as a primary opioid, a higher dose was needed as only a mu-receptor effect was aimed at. In such situations, a fre-quent dosing of 10 mg tablets up to q4-5h for the first few days could be required, usually fol-lowed by three times daily [39]. Earlier reporting has described how the duration of analgesia produced by intravenous morphine or methadone during the first few days did not differ sig-nificantly [39, 40], whereas there is a marked difference in long-term treatment.

Severe pain is a powerful stimulant for the respiratory center of the brain stem [41, 42]. For this reason, opioid doses can be rapidly escalated in cases with insufficient opioid sensitivity, with little risk for respiratory depression. However, if the pain is suddenly relieved by other means, e.g. by a nerve block—or by a successful addition of methadone—the stimulus on the brain stem diminishes, resulting in potential respiratory depression. For this reason, experi-enced physicians preferred to decrease the dose of the basic opioid, rather than the low dose of methadone.

According to several physicians, the duration of the analgesic effect of low dose methadone was relatively short, from only a few weeks to a few months, whereas the effect of methadone as a primary opioid was claimed to be much longer. However, with the usually short expected survival of patients in specialized palliative care, this is of secondary importance. Thus, differ-ent perspectives on the need for long-term efficacy, e.g. between palliative and pain medicine,

could also explain the reported different views on the benefits of low-dose add-on therapy with methadone.

Already in 2011, McKenna et al. described how the particular use of low-dose add-on methadone could be advantageous in an outpatient palliative care setting, provided there was a close review of the patient and well-informed relatives. According to our informants, initiation of low-dose add-on methadone in a home care setting has become routine and provides a great advantage over having to admit the patient for rotation on a ward. This finding is in line with our previous reporting from 2020, where we described how a third of methadone initiations in specialized palliative care in Sweden were performed at home [18].

At the end of life, when oral administration becomes increasingly difficult, adequate pain relief is still required. In the hands of our informants, subcutaneous injections were often used to replace methadone tablets. Further, use of methadone together with other common drugs such as haloperidol, midazolam, metoclopramide and hyoscine butyl bromide in a subcutaneous syringe driver was described as both effective and complication-free, in line with existing data [43].

This study has several strengths, primarily, we explored not only attitudes towards methadone use for complex pain, but also its actual, practical use. Additionally, the qualitative conventional content analysis allowed an exploration of phenomena where only limited literature is currently available.

We do recognize some limitations to our study. The conclusions are based on reported practices and not on actual observations and there could be biases both towards response and recall. Moreover, results from qualitative research require that the reader, based on the description of the context, transfer relevant results and draw conclusions that apply to his or her own situation.

### Recommendations for future research

There is emerging evidence that low-dose methadone as an add-on twice daily to already existing high-dose opioid treatment increases the pain-relieving effect in complex pain [14–18]. This is also the general opinion in the current study. However, so far, this is not an evidence-based approach. Randomized controlled trials would clarify the true effect of methadone as an add-on therapy and elucidate the magnitude of the effect. Further studies on the duration of analgesia and which pain types and associated diagnoses that may benefit from low-dose add-on methadone would also be of interest.

Moreover, there is a need of separate studies with methadone as a primary opioid. Whereas the NMDA-inhibiting effect is aimed at in a low-dose setting, methadone´s mu-receptor effect is equally desirable when using methadone as a substitute for another strong opioid. In such a setting, and with that aim, both total doses and dosing might be different from the low-dose strategies. E.g., when intravenous methadone as a primary opioid was compared with morphine in a patient controlled analgesia study for 6 days, the patients pushed the button for next dose after 3.9 hours (+/- 0.85 hours) in the morphine arm and after 3.9 hours (+/- 1.6 hours) in the methadone arm [40]. Thus, whereas a two-dose regimen is recommended in the low-dose setting today, different regimens (e.g. three-dose regimens) might be optimal when methadone is used as a primary opioid. Randomized controlled trials are needed to clarify such hypotheses.

### Conclusions

There are relatively few negative attitudes towards methadone among physicians, patients or relatives in Swedish specialized palliative care. According to our informants, selected patients

may benefit from the use of methadone; either as a primary therapy or, when time is limited, unexpectedly good and safe analgesia can be achieved with low-dose add-ons to high doses of other opioids. This regimen can also be used in home care, a finding which is supported by recent studies [17, 18]. Also, patients with complex cancer-related pain involving central sensitization were considered to be the most appropriate candidates for methadone treatment.

Though associated with drug abuse, the general attitude towards methadone among palliative care physicians was positive and did not constitute an obstacle to its use. Methadone is, however, no magic bullet and one must warn against over-use with low precision.

## Supporting information

**S1 Text. Semi-structured interview guide.**
(DOCX)

## Acknowledgments

The authors would like to thank all the physicians who participated as informants for their valuable contribution. David Boniface is acknowledged for linguistic revision.

## Author Contributions

**Conceptualization:** Per Fürst, Staffan Lundström, Peter Strang.

**Data curation:** Per Fürst.

**Formal analysis:** Per Fürst, Peter Strang.

**Funding acquisition:** Peter Strang.

**Investigation:** Per Fürst, Peter Strang.

**Methodology:** Per Fürst, Staffan Lundström, Peter Strang.

**Project administration:** Per Fürst.

**Resources:** Per Fürst.

**Software:** Per Fürst.

**Supervision:** Staffan Lundström, Peter Strang.

**Validation:** Per Fürst, Peter Strang.

**Visualization:** Per Fürst.

**Writing – original draft:** Per Fürst, Peter Strang.

**Writing – review & editing:** Per Fürst, Staffan Lundström, Peter Strang.

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
