## [Decision Letter · Decision Letter 0]

12 Feb 2020

PONE-D-20-00882

Methadone in Swedish specialized palliative care – is it the magic bullet in complex cancer-related pain?

PLOS ONE

Dear Dr. Fürst,

Thank you for submitting your manuscript to PLOS ONE. After careful consideration, we feel that it has merit but does not fully meet PLOS ONE’s publication criteria as it currently stands. Therefore, we invite you to submit a revised version of the manuscript that addresses the points raised during the review process.

We would appreciate receiving your revised manuscript by Mar 28 2020 11:59PM. To enhance the reproducibility of your results, we recommend that if applicable you deposit your laboratory protocols in protocols.io, where a protocol can be assigned its own identifier (DOI) such that it can be cited independently in the future. For instructions see: http://journals.plos.org/plosone/s/submission-guidelines#loc-laboratory-protocols

We look forward to receiving your revised manuscript.

Kind regards,

Tim Luckett

Academic Editor

PLOS ONE

2. Please include additional information regarding the interview guide or script used in the study and ensure that you have provided sufficient details that others could replicate the analyses. For instance, if you developed a guide as part of this study and it is not under a copyright more restrictive than CC-BY, please include a copy, in both the original language and English, as Supporting Information.

Reviewers' comments:

Reviewer's Responses to Questions

**Comments to the Author**

1. Is the manuscript technically sound, and do the data support the conclusions?

Reviewer #1: Yes

Reviewer #2: Yes

2. Has the statistical analysis been performed appropriately and rigorously? 

Reviewer #1: N/A

Reviewer #2: N/A

3. Have the authors made all data underlying the findings in their manuscript fully available?

Reviewer #1: No

Reviewer #2: Yes

4. Is the manuscript presented in an intelligible fashion and written in standard English?

Reviewer #1: No

Reviewer #2: Yes

5. Review Comments to the Author

Reviewer #1: The manuscript is technically sound and the data supports the conclusions. The use of methadone is under studied and this qualitative study adds to the literature. The authors state " Due to the Swedish regulations based on the GDPR, and the ethical requirements

stipulated by EPN (The Regional Ethical Review Board in Stockholm Region), excerpts

of data will be only made available upon request to Dr. Per Fürst, at per.furst@ki.se,

who will forward the request to the official in charge of the GDPR restricted data at

Stockholms Sjukhem´s Research and Development unit."

There are minor grammatical errors at lines 26 (imposing-meaning unclear), 114, 166, 180 (how long an answer), 192 (other types of patients) , 193 (a limited..). At Line 394 do the authors mean muscle spasm by the term "convulsive state"? At line 366, the data as presented do not support the conclusion that the participants may overlook other cancer pain strategies in their enthusiasm for methadone. Can you please provide data for this conclusion?

Reviewer #2: I was delighted to see this exploration of methadone use in palliative care. Qualitative pieces have utility in this field, many of us have the perception that negative attitudes (due to stigma, etc) have played a large part in preventing widespread use of methadone at end-of-life.

This is a strong piece and the majority of my comments are grammatical, rather than technical. This is, I'm aware, outside of my scope as a peer reviewer; I include them here only because PlosOne does not employ an editor and part of my instructions were to correct typographical errors:

1. End-of-life does not need to be capitalized.

2. Lines 25-26 are unclear. I believe that you are saying that physicians were surprised by how little stigma they encountered, but I am not sure what "not imposing methadone's analgesic use" means in this context.

3. You are missing a comma at the end of line 26.

4. Instead of "low dose add-on to other opiates," you may want to describe this method as "adjuvant methadone;" I believe that most of your readership will be familiar with this parlance, especially if you define it once early in the piece

5. I am a little unclear on how your participant selection worked. It seems like you searched from a list of palliative care physicians, and then aimed to select as diverse a group as possible? If so, who generated this list? Was there some level of snowball sampling at work here?

6. Lines 149-150: Did you mean to say that the initial focus "was to explore NEGATIVE attitudes and opinions"? If not, I don't understand the next sentence, which implies that the opinions were surprisingly positive.

7. Line 157: I'm not sure what you mean by "delineations of situations." Perhaps it would be better to just say "situations" here

8. It may be worth noting that although people perceive methadone to be most effective in managing neuropathic pain, the little evidence we have is inconclusive (https://www.cochrane.org/CD012499/SYMPT_methadone-neuropathic-pain-adults)

9. Lines 351-352: I recommend a semicolon here, rather than a period

Overall, a very interesting piece exploring prescriber perceptions.

6. PLOS authors have the option to publish the peer review history of their article (what does this mean?). If published, this will include your full peer review and any attached files.

Reviewer #1: No

Reviewer #2: No

---

## [Author Response · Author response to Decision Letter 0]

20 Feb 2020

Response to Reviewers, PONE-D-20-00882

Methadone in Swedish specialized palliative care – is it the magic bullet in complex cancer-related pain?

Dear Editor, 

Thank you for your encouraging decision letter. Please find our response below.

After this revision, we hope the manuscript will be suitable for publication in PLOS ONE. 

With kind regards

Per Fürst

MD, Consultant in palliative medicine

Karolinska Institutet and Stockholm Sjukhem Foundation

All changes are marked in yellow in the revised manuscript. 

Response to Academic Editor and journal requirements:

1. Based on the style requirements by PLOS ONE we have reviewed the manuscript, including the file naming. 

2. We have included as Supporting Information, named S1 Text, a copy of the interview guide in English and Swedish.

3. As researchers we have the utmost respect for the value of publishing scientific data as a crucial factor for increased transparency in the research process. Unfortunately, in the context of this study, there are some important limitations that prevent us from publishing the qualitative interviews on-line.

When planning this study, an approval from Regional Ethical Review Board of Stockholm (EPN) was an absolute requirement. According to the requirements of EPN (Dnr. 2017/2302), a finalized protocol of informed consent was to be included in the application. The exact wording in this protocol/letter of consent was “…all results will be presented anonymously and in the form of processed and condensed data in a scientific paper. It will not be possible to identify individual persons from the material. Your personal data will be handled according to “Personuppgiftslagen” (Privacy Act) PUL 1998.204…” These regulations have been strengthened further with the introduction of GDPR in Sweden (2019). For this reason, we are not able to publish such data on-line.

Further, the data we have collected and used as the basis for our analysis in this study are interviews with individual informants, all of whom are clinically active physicians in specialized palliative care in various locations in Sweden. The number of physicians in this medical area is limited and if you are an active physician or other healthcare professional in the area you can probably in many cases read through and, with fairly high accuracy, guess who the informants are. All in all, it is therefore our assessment that even after fairly extensive de-identification, the risk that the informants' identities could be revealed is too great to allow the interviews to be published on-line.

To try to meet the scientific requirements we therefore provide contact information to the Data Protection Representative responsible for GDPR at Stockholm Sjukhem, e-mail dso@stockholmssjukhem.se to whom anyone can turn in case of a further request. 

Response to Reviewer 1’s comments 

Thank you for your valuable comments to which we respond below. 

1. Grammatical errors are corrected as follows: 

o Line 26 now 29. The word “imposing” has been changed to “affecting methadone’s use as an analgesic”.

o Line 114 now 117. The comma after “[21)” was removed and the first letter after the colon sign was capitalized.

o Line 166 now 169. “a low precision” was changed to “low precision”

o Line 180 now line 187. The word “an” was added.

o Line 192 now line 199. Changed to “types”

o Line 193 now line 200. “a limited” changed to “limited”

o Line 394 now line 403. “a convulsive state” changed to “muscle spasm”

2. Thank you for requesting further presentation of data to prove the conclusion that the participants may overlook other cancer pain strategies in their enthusiasm for methadone (line 366, now line 371). We find that this improves the manuscript and data is now added in the form of a new citation in the Findings section (line 172): 

[Pain specialist:] When methadone is introduced, it sometimes implies an overuse of an unconsidered treatment, while there probably sometimes are [alternative] oncological treatments, or it is more proper to consider a spinal catheter or something else. (Informant 14)

Response to Reviewer 2’s comments 

Thank you for your valuable comments. Please see below for our response. 

1. Capitalizations of “end-of-life” are removed throughout the manuscript.

2. Line 26 now 29. The word “imposing” has been changed to “affecting methadone’s use as an analgesic”.

3. Line 26 now line 30. “bone metastatic” is changed to “bone-metastatic”.

4. Thank you for this suggestion. As non-native English speakers we have carefully considered whether we should use the words low-dose add-on, coanalgesic or adjuvant to describe the use of low-dose methadone in order to achieve an analgesic effect that may appear, at least occasionally, when a low dose of methadone is added to an already ongoing opioid therapy. We have concluded that all three terms have a similar, but not identical, meaning. We have also found that the definition of "adjuvant analgesics" primarily refers to the use of drugs that have a primary indication that is other than pain (see e.g. Lussier D. Oncologist 2004. Adjuvant analgesics in cancer pain management). When it comes to methadone, there are primarily two different pain-relieving effects that we are likely to benefit from, the my-receptor stimulant effect and the NMDA-receptor inhibiting effect. Instead, we have therefore considered using the word coanalgesic, which occurs elsewhere in the literature (e.g. Courtemanche F. Methadone as a Coanalgesic for Palliative Care Cancer Patients. J of Pall Med 2016), but eventually concluded that we wish to keep the term low-dose add-on as we think it distinguishes low-dose therapy from regular or high-dose methadone therapy when used a primary analgesic.

5. Regarding sampling methods. Specialized palliative care in Sweden is an area with a limited number of active physicians. At present, there are about 100 physicians who are specialists in palliative medicine and, additionally also about 350 other physicians, also members of SFPM (Swedish Association of Palliative Medicine). We have at least superficial knowledge of each other in this community and it was therefore from this knowledge that we were able to carry out a max variation sampling at a national level. Snowball sampling was not used in this study, since snowball sampling, according to Patton (In, Patton M Q. “Qualitative Research & Evaluation Methods”, p. 237) is especially useful when a study focuses on information-rich key informants or critical cases. Our aim was to catch a broad description of the phenomenon, as we partly focused on attitudes. Therefore, we looked for both experienced and less experienced informants. 

6. Lines 149-151 now lines 152-155. Thank you for directing our attention to this incongruency in the manuscript. The full paragraph was changed as follows: 

“The initial focus of the study was to explore negative as well as positive attitudes and opinions regarding methadone. However, it soon became apparent that they negative attitudes were few among physicians. In fact, most of the informants were more enthusiastic about, and provided spontaneous aspects on, the benefits and practical uses of methadone.”

7. Line 157 now 160. The words “delineations of” were removed. 

8. We have added a sentence in the Attitudes section of the Discussion which highlights the important conclusions of the 2017 Cochrane’s review on methadone and neuropathic pain. 

9. Line 351-532 now line 358. A semicolon was added.

---

## [Decision Letter · Decision Letter 1]

6 Mar 2020

PONE-D-20-00882R1

Methadone in Swedish specialized palliative care – is it the magic bullet in complex cancer-related pain?

PLOS ONE

Dear Dr. Fürst,

On the whole, you have done an excellent job of addressing the reviewers' comments, and I can confirm that your manuscript is very nearly ready for publication. However, please just add some recommendations for future research after your limitations section.

We would appreciate receiving your revised manuscript by Apr 20 2020 11:59PM. To enhance the reproducibility of your results, we recommend that if applicable you deposit your laboratory protocols in protocols.io, where a protocol can be assigned its own identifier (DOI) such that it can be cited independently in the future. For instructions see: http://journals.plos.org/plosone/s/submission-guidelines#loc-laboratory-protocols

We look forward to receiving your revised manuscript.

Kind regards,

Tim Luckett

Academic Editor

PLOS ONE

Reviewers' comments:

Reviewer's Responses to Questions

**Comments to the Author**

1. If the authors have adequately addressed your comments raised in a previous round of review and you feel that this manuscript is now acceptable for publication, you may indicate that here to bypass the “Comments to the Author” section, enter your conflict of interest statement in the “Confidential to Editor” section, and submit your "Accept" recommendation.

Reviewer #1: All comments have been addressed

Reviewer #2: All comments have been addressed

2. Is the manuscript technically sound, and do the data support the conclusions?

Reviewer #1: Yes

Reviewer #2: Yes

3. Has the statistical analysis been performed appropriately and rigorously? 

Reviewer #1: Yes

Reviewer #2: Yes

4. Have the authors made all data underlying the findings in their manuscript fully available?

Reviewer #1: No

Reviewer #2: Yes

5. Is the manuscript presented in an intelligible fashion and written in standard English?

Reviewer #1: Yes

Reviewer #2: Yes

6. Review Comments to the Author

Reviewer #1: Thank you for addressing the comments raised. The only addition which would strengthen the paper would be recommendations for further research.

Reviewer #2: Thank you for your revisions. This piece is a valuable contribution to the field of palliative care.

7. PLOS authors have the option to publish the peer review history of their article (what does this mean?). If published, this will include your full peer review and any attached files.

Reviewer #1: No

Reviewer #2: No

---

## [Author Response · Author response to Decision Letter 1]

9 Mar 2020

Response to Reviewers, PONE-D-20-00882

Methadone in Swedish specialized palliative care – is it the magic bullet in complex cancer-related pain?

Dear Editor, 

Thank you for your decision letter. Please find our response below.

With kind regards

Per Fürst

MD, Consultant in palliative medicine

Karolinska Institutet and Stockholm Sjukhem Foundation

All changes are marked in yellow in the revised manuscript with track changes. 

Response to Reviewer 1’s comments 

Thank you for your valuable comment!

After our limitations section we now have added, in lines 463-479, a new section with the subheading “Recommendations for future research”. 

“There is emerging evidence that low-dose methadone as an add-on twice daily to already existing high-dose opioid treatment increases the pain-relieving effect in complex pain [14-18]. This is also the general opinion in the current study. However, so far, this is not an evidence-based approach. Randomized controlled trials would clarify the true effect of methadone as an add-on therapy and elucidate the magnitude of the effect. Further studies on the duration of analgesia and which pain types and associated diagnoses that may benefit from low-dose add-on methadone would also be of interest.

Moreover, there is a need of separate studies with methadone as a primary opioid. Whereas the NMDA-inhibiting effect is aimed at in a low-dose setting, methadone´s mu-receptor effect is equally desirable when using methadone as a substitute for another strong opioid. In such a setting, and with that aim, both total doses and dosing might be different from the low-dose strategies. E.g., when intravenous methadone as a primary opioid was compared with morphine in a patient controlled analgesia study for 6 days, the patients pushed the button for next dose after 3.9 hours (+/- 0.85 hours) in the morphine arm and after 3.9 hours (+/- 1.6 hours) in the methadone arm [40]. Thus, whereas a two-dose regimen is recommended in the low-dose setting today, different regimens (e.g. three-dose regimens) might be optimal when methadone is used as a primary opioid. Randomized controlled trials are needed to clarify such hypotheses.”

---

## [Editor Report · Decision Letter 2]

11 Mar 2020

Methadone in Swedish specialized palliative care – is it the magic bullet in complex cancer-related pain?

PONE-D-20-00882R2

Dear Dr. Fürst,

We are pleased to inform you that your manuscript has been judged scientifically suitable for publication and will be formally accepted for publication once it complies with all outstanding technical requirements.

With kind regards,

Tim Luckett

Academic Editor

PLOS ONE

---

## [Editor Report · Acceptance letter]

13 Mar 2020

PONE-D-20-00882R2 

Methadone in Swedish specialized palliative care – is it the magic bullet in complex cancer-related pain? 

Dear Dr. Fürst:

I am pleased to inform you that your manuscript has been deemed suitable for publication in PLOS ONE. Congratulations! Your manuscript is now with our production department. 

With kind regards,

on behalf of

Dr. Tim Luckett 

Academic Editor

PLOS ONE